# Limited Budget Adversarial Attack Against Online Image Stream

**Hossein Mohasel Arjomandi** [1]   **Mohammad Khalooei** [1]   **Maryam Amirmazlaghani** [1]

## Abstract

An adversary wants to attack a limited number of images within a stream of known length to reduce the exposure risk. Also, the adversary wants to maximize the success rate of the performed attacks. We show that with very minimal changes in images data, majority of attacking attempt would fail, however some attempts still lead to succeed. We detail an algorithm that choose the optimal images which lead to successful attack . We apply our approach on MNIST and prove it's significant outcome compared to the state of the art.

## 1. Introduction

We are interested in an adversarial attack against an image stream of known length $N$. In an image stream, the sender transmits a sequence of $N$ images to the receiver. The receiver receives each image and classifies it. An adversary conducts a man-in-the-middle attack and accesses each image upon arrival from the sender and intends to attack a maximum of $k$ images among whole $N$ images, then send them to the receiver so that the receiver misclassifies them. Other images that are not chosen will be passed to the receiver untouched. The target model, which is the receiver's model, is a deep neural network with fixed weights. The adversary is interested in maximizing its fool rate, which is the number of successfully performed attacks over $k$. We assume $k \ll N$ because the adversary wants to minimize its footprint and the risk of being detected. The receiver knows the approximate frequency of the image arrival and their order, So, upon the arrival of each image, the adversary must irrevocably decide whether attack the current image or wait for the future image which could yield a higher misclassification probability than the current image. Any image received by the adversary should be sent to the receiver before the next image arrives. The adversary finds out the success or failure result only after the stream ended.

*Equal contribution   [1]Department of Computer engineering, Amirkabir University of Technology, Tehran, Iran. Correspondence to: h.mohassel,khalooei,mazlaghani <@aut.ac.ir>.

*Accepted by the ICML 2021 workshop on A Blessing in Disguise: The Prospects and Perils of Adversarial Machine Learning.* Copyright 2021 by the author(s).

## 2. Literature Review

Few papers addressed attacking sequential data. (Lin et al., 2017) created a reinforcement learning (RL) agent that attacked in limited states of a sequence of states and used it against Atari player agents. (Gong et al., 2019) devised a RL agent that observed data to create perturbation for unobserved data within a stream. (Sun et al., 2020) created a RL agent that could predict the target agent's action, find the critical point of attacking it, and tested their approach against game player agents. (Mladenovic et al., 2021) addressed the same problem as ours, but we explain its approach is erroneous. (Guo et al., 2019) implemented a simple and fast black box attack with few queries. (Ilyas et al., 2018) used tiling technique to reduce the input dimension. These papers emphasize the importance of query-efficient black box attack methods. In the following sections, we introduce our threat model, criteria algorithm, and our proposed black box attack method, and our results.

## 3. Threat Model

We define an adversarial example as:

$$\|p\|_\infty \leq \epsilon \tag{1}$$

$$x' = x + p \tag{2}$$

$$F(x') \neq y \tag{3}$$

In the above formulas, $x$, $y$, $p$ and $x'$ are a clean image, its ground truth label, the perturbation and the perturbed image coresponding to the target classsifier function $F$, respectively. We address both black and white box methods. In white box approach, selecting the optimal images to attack is trivial since the adversary knows if the current image yields a successful attack. The surrogate model black box approach looks a similar case. If the surrogate model is good enough that preserves output logit ordering, the adversary could already know the result of each attack. We also suppose the adversary knows the number of classes, $m$.

Nevertheless, we use the *Fast Gradient Sign Method (FGSM)* of (Goodfellow et al., 2014) in our experiments which we will explain later. In our black box setup, the adversary has unlimited query access to the target model and can get the classification loss of the query. We assume that the target model uses *Cross Entropy (CE)* loss for the

classification and that the target model does not use any form of regularization. For an image $x$ we define:

$$loss(x, y) = CE(F(x), y) = -\ln(p(y)) \qquad (4)$$

## 4. Criteria Algorithm

### 4.1. Grouping Images by Loss

Since the adversary has access to loss in both white and black box scenarios and no regularization loss exists, the adversary can find the probability of ground truth class after each query by computing as $p(y) = e^{-CE(F(x'))}$. This information helps to define three loss groups:

**Low-loss group (LLG)** : Images in this group will have $e^{-CE((F(x'))} > 1/2$. These images will lead to failed attack since no more than one class could have a predicted probability of greater than 0.5. These images will never be chosen.

**High-loss group (HLG)** : Images in this group will have $e^{-CE((F(x'))} < 1/m$. Regarding the pigeonhole principle, given $m$ real numbers whose sum is 1, at least one of them is not less than their average, the $1/m$. So after the HLG images are attacked, the ground truth class probability is no longer dominant and will result in a successful attack. Images in HLG will be chosen for attack if encountered.

**Medium-loss group (MLG)** : Images belonging to none of the two above groups fall into MLG, and the adversary could not find if the attack will be successful or not and must use a criteria algorithm which will be described in section 4.2.

### 4.2. Secretary Algorithm for Medium-Loss Group

If the adversary rejects the arriving LLG data, it will no longer know $n$, the size of the non-LLG images of the stream and only knows $N$, the initial stream size, which is the upper bound on $n$. We assume the distribution of $n$ on set $\{1, 2, \ldots, N\}$ is uniform since the adversary has no knowledge about the target model robustness against the current $\epsilon$.

Although the adversary is only interested in maximizing its fool rate, when the stream is ongoing, there is no way it can find whether an image in MLG leads to a successful attack or not. In order to tackle this problem, the adversary needs to associate each incoming image to a metric, which is a real number. The higher metric should reflect the higher chance of MLG image leading to a successful attack. The metric will be discussed in section 4.3. So, after defining a metric, the adversary must adopt a criteria algorithm to maximize the total metrics values of the images selected from MLG.

The adversary does not know anything about future images' metrics and can only remember visited ones. This problem is the k-secretary problem of unknown queue length, where $k$ is the number of elements to be chosen from the queue.

The k-secretary problem with the queue of an unknown length concerns irrevocably choosing $k$ elements of a queue of unknown length to maximize the metric values associated with the selected elements. Nothing about the distribution of elements' metrics is known. The general case for $k$ is not addressed in the literature yet, but for $k = 1$, assuming the distribution of $n$ on set $\{1, 2, \ldots, N\}$ is uniform (Presman, 1973) showed the optimal selector approach. In this paper, the adversary, should undergo two phases: Observation and selection phases. In the observation phase, the adversary should only observe elements up to the index $t = \lfloor N/e^2 \rfloor$, rejecting all observed elements and only remembering the metric value of the best element rejected. In the selection phase, from index $t + 1$ to $n$, it should choose the first element whose metric is bigger than the highest metric visited in the observation phase. Since $k \ll N$, we use the results of the $k = 1$ case and the VIRTUAL+ algorithm proposed in (Mladenovic et al., 2021). That is, in the observation phase, from index $t = 1$ to index $\lfloor N/e^2 \rfloor$, the adversary only observes all MLG images metrics and remembers a list $R$ of length $k$ of the highest MLG metrics visited so far and ignores all observed images. In the selection phase, which is indices $t + 1$ to $n$, the current image will be chosen if its metric is bigger than the lowest metric value of $R$; if the current image is chosen, the adversary remembers its metric value, forgetting the lowest metric value of $R$. If the adversary observed HLG data in any phase, it should attack it and reduce the $k$ to $k - 1$, and move on. If it observed LLG data, it should ignore it and continue. If the number of unseen data equals the remained choices, all non-LLG data must be chosen. Algorithm 2 presents our criteria algorithm.

### 4.3. Metrics

We study three metrics and compare results using FGSM.

**After-attack loss**: For a clean image $x$, this metric is $CE(F(x'), y)$ where $x'$ and $y$ are the perturbed image and its real label, respectively. The reason for this metric is that higher loss means lower probability of the ground truth label and higher probability of misclassification.

**After and before attack loss difference**:For a clean image $x$, this metric is $CE(F(x'), y) - CE(F(x), y)$. The reason for this metric is that a higher loss difference means a more robust attack and a higher probability that the ground truth label is no longer dominant.

$L_2$ **gradient norm of clean image**: For a clean image $x$, this metric is $\|\partial CE(F(x), y)/\partial x\|_2$. The idea behind this

metric is that regarding Taylor's series for small $\epsilon$ we have:

$$loss(x + \epsilon, y) \approx loss(x, y) + \frac{\partial loss(x,y)}{\partial x} \times \epsilon \quad (5)$$

So, a steeper gradient will probably result in a higher loss if the gradient path is followed. So far, we have used FGSM, a white box approach, to compare three metrics. Referring to Table 2, we have chosen after-attack loss as our metric of choice. Now it is time to tackle the original non-trivial problem, which is a black box attack. In section 4.4, we propose our black box attack method.

### 4.4. The Back Box Attack Method

To tackle our original non-trivial problem of image selection in a black box query-based attack, we need to propose a black box attack method for computing the perturbation and the after-attack loss metric. Inspired by (Ilyas et al., 2018; Guo et al., 2019), we have defined the *Fast Black Box attack (FBB)* method. We have used the tiling idea, which is partitioning the input image into non-overlapping tiles. We have assumed given that tiles' dimensions are small, pixels within each tile correspond to the same object in an image; thus, the gradient of all pixels within a tile are approximately constant. Also, according to 5, we have supposed the largest loss happens on the boundary of the $l_\infty$ ball with a radius of $\epsilon$. Algorithm 1 indicates the pseudocode of this attack.

---

**Algorithm 1** FBB Attack

---

  **Input:** clean image $x$ and its label $y$, tiles dimension $t$
  **Output:** $P$, the calculated perturbation for $x$ (Divide $x$ into $t \times t$ pixel tiles)
  Let $P = 0$ be the crafted perturbation of shape $x$
  **for** tile $T$ in $x$ **do**
    $x_{plus} = x; x_{minus} = x$
    $x_{plus}[T] = x_{plus}[T] + \epsilon; x_{minus}[T] = x_{minus}[T] - \epsilon$
    **if** $loss(x_{plus}, y) > loss(x_{minus}, y)$ **then**
      $P[T] = \epsilon; x = x_{plus}$
    **else**
      $P[T] = -\epsilon; x = x_{minus}$
    **end if**
  **end for**
  return $P$

---

## 5. Experimental Results and Discussion

For a convolutional neural network model, we have tested our approach on MNIST test set. To get the results of Table 2, we have analyzed numerous random permutations of the full MNIST test set, 1000 in our experiment, for all three metrics and performed our criteria algorithm with each metric separately.

---

**Algorithm 2** Criteria Algorithm

---

  **Input:** stream parameters$(X, N, k)$
  Let $R = [], i = 1, j = \lfloor N/e^2 \rfloor + 1, s = 0, r = k$
  **while** $i < \lfloor N/e^2 \rfloor$ and $s < k$ **do**
    **if** $X[i] \in$ low-loss **then**
      continue
    **else if** $X[i] \in$ high-loss **then**
      Attack $X[i]; s = s + 1; r = r - 1$
      **while** $len(R) > r$ **do**
        $R.remove(R[0])$
      **end while**
    **else if** $len(R) < r$ **then**
      $R.append(Metric(X[i]); R.sort()$ //Ascending
    **else if** $Metric(X[i]) > R[0]$ **then**
      $R[0] = Metric(X[i]); R.sort()$ //Ascending
    **end if**
    $i = i + 1$
  **end while**
  **while** $j \leq N$ and $s < k$ **do**
    **while** $len(R) > r$ **do**
      R.remove$(R[0])$
    **end while**
    **if** $X[j] \in$ low-loss **then**
      continue
    **else if** $X[j] \in$ high-loss **then**
      Attack $X[j]; s = s + 1; r = r - 1$
    **else if** $Metric(X[j]) > R[0]$ **then**
      $R[0] = Metric(X[j]);$
      Attack $X[j]; s = s + 1;$R.sort() //Ascending
    **else if** $N - j + 1 \leq r$ **then**
      **for** $q = j$ to $N$ **do**
        **if** $X[q] \notin$ low-loss **then**
          Attack $X[q]; s = s + 1;$
        **end if**
      **end for**
      break
    **end if**
  **end while**

---

For the black box case, we have also implemented our FBB method on 1000 random permutations of full MNIST test set for various $\epsilon$ and obtained the Table 1 results. Regarding both tables, the LLG size and HLG size columns show the length of these two groups among all data. The next two columns show the number of data chosen by the secretary and from HLG, respectively. Then comes the success rate for the secretary chosen data in the next column. After that, the random selection success rate in MLG is presented. The next column shows the random selection success rate among non-LLG data, which is the case where $m$ is not known. The last column is the percentage of chosen data divided by $k$. If $\epsilon$ is too low, there will not be enough non-LLG data and fewer than $k$ images will be chosen, leading to a low fool

*Table 1.* FBB attack results on full MNIST test set with $4 \times 4$ tiles averaged for 1000 random permutation with after attack loss metric. (RS-SR,F-R and N-C stand for random selection success rate, fool-rate and number of chosen data)

| $k$ | $\epsilon$ | AVERAGE F-R (%) | LLG SIZE | HLG SIZE | N-C FROM HLG | N-C BY SECRETARY | SECRETARY S-R (%) | RS-SR IN MLG | RS-SR NOT KNOWING $m$ | $\frac{\text{N-C}}{k}$ (%) |
|---|---|---|---|---|---|---|---|---|---|---|
| 1000 | | 35.4 | | | 250.4 | 106 | **98.5** | | | 35.6 |
| 100 | $\frac{1}{20}$ | **98.9** | 9353 | 251 | 68.5 | 31.5 | **98.1** | **89.1** | **93.3** | **100** |
| 10 | | **100** | | | 10 | 0 | - | | | **100** |
| 1000 | | 74.4 | | | 516 | 231.4 | **98.5** | | | 74.7 |
| 100 | $\frac{3}{40}$ | **100** | 8632 | 517 | 90.7 | 9.3 | **100** | **88.4** | **92.8** | **100** |
| 10 | | **100** | | | 10 | 0 | - | | | **100** |
| 1000 | | **99.6** | | | 687.1 | 312.9 | 98.7 | | | **100** |
| 100 | $\frac{1}{10}$ | **100** | 7344 | 1117 | 100 | 0 | - | **93.9** | **90.2** | **100** |
| 10 | | **100** | | | 10 | 0 | - | | | **100** |

*Table 2.* Comparison of metrics in FGSM attack with $\epsilon = 0.05$ on full MNIST test set averaged for 1000 random permutation. (RS-SR,F-R and N-C stand for random selection success rate, fool-rate and number of chosen data)

| METRIC | $k$ | AVERAGE F-R (%) | LLG SIZE | HLG SIZE | N-C FROM HLG | N-C BY SECRETARY | SECRETARY S-R (%) | RS-SR IN MLG | RS-SR NOT KNOWING $m$ | $\frac{\text{N-C}}{k}$ (%) |
|---|---|---|---|---|---|---|---|---|---|---|
| | 1000 | 58 | | | 410.8 | 168.8 | **98.5** | | | 58 |
| $\triangle CE$ | 100 | **100** | | | 83.4 | 16.6 | **98.9** | | | **100** |
| | 10 | **100** | | | 10 | 0 | - | | | **100** |
| | 1000 | 57 | 8968 | 412 | 411 | 168.2 | 92.3 | 87.3 | 92.4 | 57.9 |
| $l_2$ GRAD | 100 | 98 | | | 83 | 16.9 | 90.8 | | | **100** |
| | 10 | **100** | | | 10 | 0 | - | | | **100** |
| | 1000 | 58 | | | 410.7 | 169 | **98.4** | | | 58 |
| $CE(x')$ | 100 | **100** | | | 83.1 | 16.9 | **98.9** | | | **100** |
| | 10 | **100** | | | 10 | 0 | - | | | **100** |

rate. A set's success rate is the number of images in the set leading to a successful attack divided by the set size.

As Table 2 shows, the after-attack loss and the loss difference metrics yield the same fool rate, but since the former uses half the queries of the latter, it was the metric of choice in our black box approach. In contrast, the $L_2$ gradient norm metric has not performed higher than random selection in the non-LLG data when not knowing $m$, so it is not a good metric ($m$ is equal to 10 in *MNIST* dataset (LeCun & Cortes, 2010)). Regarding Table 1 and Table 2, even relatively low epsilons lead to high fool rates. There are several reasons for this. First, we only need a few successful attacks to reach high fool rates. Second, even with low epsilons, many images belong to HLG , and for sufficiently large $k$ such as k=100 and k=1000 cases in both tables, our algorithm almost selects all HLG images. Third, since the vast majority of images belong to LLG and the vast majority of MLG lead to a successful attack, the success rate among the non-LLG group is high and the adversary still gets above 90% fool rates only by rejecting LLG data and choosing randomly among non-LLG data, that is, MLG and HLG combined. Also, as Table 1 and Table 2 suggest, introduc-

ing the secretary approach slightly improves the random selection strategy among MLG. Tiles' dimension is another important parameter. Should the adversary have enough time between image arrivals, smaller tiles are preferable because they allow greater exploitation of the adversarial space, thus; requiring lower epsilons.

Comparing our Table 1 results with (Mladenovic et al., 2021), with $\epsilon = 3/40$ which is one-fourth of their $\epsilon$, we outperformed their white box approach on all $k$ equal to 10,100 and 1000. Typically, epsilons between 0.2 and 0.3 are used for MNIST. We do not need these high epsilons because we are interested in fooling a small share of the stream. It should be noted that although the adversary knows the initial length of the stream, $N$, the vast majority of images are rejected because they belong to LLG; thus, we used the secretary problem with the unknown length of the stream.

Note that when $k$ is relatively large, the $k = 1000$ case, the low fool rate is because fewer than $k$ images belong to non-LLG. We tackled this issue by switching to slightly higher epsilons, 3/40 and 1/10 namely. .

## 6. Conclusion

Our approach results in high fool rates using very low epsilons in irrevocable limited-budget online adversarial attacks with known stream length is. By introducing the loss grouping idea, fool rates significantly improved. Future work should address the theoretical rationale behind metrics and criteria algorithm, and address the general architecture with possible regularization loss.

## 7. Acknowledgement

We would like to take this opportunity to acknowledge the time and effort devoted by reviewers to improving the quality of this manuscript.

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
