# OpenReview forum: "Limited Budget Adversarial Attack Against Online Image Stream"
_ICML.cc/2021/Workshop/AML — ICML 2021 Workshop AML Poster_

### Official Review · Reviewer_SgwN · 2021-06-19
**This work proposes a criteria algorithm to choose a limited number of images from a stream to attack.**

**Rating:** Accept
**Confidence:** 4

**Review:**

An adversary wants to attack a limited number of images within a stream of known length to reduce the exposure risk. Also, the adversary wants to maximize the success rate of attacks. This work proposes a criteria algorithm to choose a limited number of images to attack.

Pros: The problem formulation is reasonable. The proposed selection criteria is reasonable.

Cons: More details about your proposed method should be included in the abstract. The writing should be improved.

---

### Decision · Program_Chairs · 2021-06-21

**Decision:**

Accept (Poster)

**Comment:**

This paper proposed a criteria algorithm to choose a limited number of images to attack. The writing can be improved.